# The experience of living with human immunodeficiency virus among adolescents at Felege Hiwot Comprehensive Specialized Hospital Bahir-Dar, Northwest Ethiopia, A phenomenological study

**Bazezew Asfaw Guadie** [1]*, **Minyichil Birhanu Belete** [2], **Meseret Mekuriaw Beyene**[1], **Ayalew Kassie Melese**[1], **Tarik Yenew Tessema**[3], **Yinager Workineh**[2]

**1** Department of Nursing Bahir Dar Health Science College, Bahir Dar, Ethiopia, **2** Department of Pediatrics and Child Health Nursing, School of Health Sciences, College of Medicine and Health Sciences, Bahir Dar University, Bahir Dar, Ethiopia, **3** Infectious Disease Screening, Bahir Dar Blood Bank Office, Amhara National Regional State Health Bureau, Bahir Dar, Ethiopia

* bazezewasfaw@gmail.com

## Abstract

### Background

Adolescents who have the human immunodeficiency virus face difficulties in their lives not just from the physical consequences of the illness but also from social stigma and discrimination. The quantitative side of this issue was the focus of earlier Ethiopian research. However, there hasn't been any prior research done extensively in Ethiopia on the real-life experiences of teenagers infected with HIV. Therefore, in order to address the real issues that these teenagers encounter, it is imperative that we investigate their lived experiences as HIV-positive adolescents.

### Objective

The goal is to investigate the experiences that adolescents at Felege Hiwot Comprehensive Specialized Hospital who are infected with HIV have on a daily basis.

### Method

A phenomenological approach was conducted among thirteen purposively selected adolescents at Felege Hiwot Comprehensive Specialized Hospital, Bahir-Dar, northwest Ethiopia, from March 25 to April 30, 2023. Information saturation was used to estimate the sample size. Data was gathered through semi-structured open-ended interview guides and in-depth interview techniques. The talk was recorded using an audio recorder. The data was transcribed verbatim. The conceptual translation approach was used to translate the transcribed data from Amharic to English. For additional analysis, the translated data was entered into the Atlas.ti.8 program. To demonstrate findings, the inductive thematic analysis technique was applied using illustrative quotes. Colleague feedback, member checks, and a debriefing were used to guarantee the quality of the data.

**Data Availability Statement:** All relevant data are within the paper and its Supporting Information files.

**Funding:** The author(s) received no specific funding for this work.

**Competing interests:** The authors have declared that no competing interests exist.

## Result

Eleven subthemes, including main theme of "lived with burdens of immunodeficiency virus," "disclosure," "ART adherence," and "future aspiration," were used to describe the findings. The participants discussed how difficult it has been for them to deal with social, emotional, and psychological difficulties in addition to living with the HIV infection. They kept their illness a secret from others out of fear of prejudice and stigma. **In conclusion,** adolescents infected with the human immunodeficiency virus faced a variety of difficulties related to their mental health, relationships, emotions, and compliance. It is advised that the community get ongoing, comprehensive health education in order to prevent stigmatization and discrimination against young people who have the virus.

## Introduction

There are around 1.2 billion teenagers worldwide, with more than 80% of them residing in poor nations [1]. HIV continues to be one of the biggest risks to global health in the modern era [2, 3]. A UNICEF research estimates that in 2015, two million teenagers globally were projected to be living with HIV, and that around 26 adolescents became infected with HIV every hour [4]. In sub-Saharan Africa, acquired immunodeficiency sickness syndrome (AIDS) was the primary cause of death for teenagers [5]. In low-middle-income nations, HIV/AIDS continues to rank among the top 10 causes of death [6]. HIV remains a public health concern in the Amhara Regional state, where youth (adolescents and young adults) aged 15 to 24 make up approximately 22% of the country's population and account for 0.73% of HIV cases in Ethiopia's urban areas [7]. The annual HIV incidence has been fluctuating, with Dessie town, Bahir Dar city, and Gondar town having the highest incidence [8].

People who are living with HIV/AIDS (PLWH) describe their lived experience as the challenges and difficulties they have faced throughout their lives as a result of contracting HIV [9]. People with HIV/AIDS encountered a variety of obstacles and difficulties, including information-seeking, cognitive, physical, psychological, and social. People with HIV experienced challenges in their daily lives, including emotional shock, fear of the repercussions, remorse, discouragement, and a desire to escape reality [5]. For example, among the negative experiences that result in social isolation and hatred after receiving an HIV diagnosis is emotional shock [5]. The period of life between childhood and adulthood, known as adolescence, lasts from the ages of 10 to 19 [5]. It is a special moment in human development and a crucial period for establishing the groundwork for long-term good health. An rising number of HIV-positive individuals worldwide are adolescents and young adults [6]. Although effective combination antiretroviral therapy (ART) has increased life expectancy for people with HIV, cognitive disorders continue to be a major concern for PLWH and those who provide care for them [10]. People with AIDS illnesses experience uncertainty about their prognoses, worse treatment outcomes, poor social relationships, identity crises, and a lower quality of life. Quality of life is recognized as an essential endpoint in the disease management of HIV/AIDS with calls to include good quality of life as a 'fourth 90' in the 90-90-90 testing and treatment targets introduced by the World Health Organization (WHO) in 2016 for people living with HIV [11].

Individuals living with HIV, especially teenagers, frequently suffer from cognitive deficits that have a variety of detrimental consequences. Despite the fact that individuals with HIV

now live longer because to excellent combination antiretroviral therapy (ART), cognitive impairments remain a serious worry for people living with HIV and those who care for them [10]. Uncertainty about their prognoses, poorer treatment results, strained social interactions, identity conflicts, and a reduced standard of living are all experienced by those with AIDS illnesses. In addition, they face social and psychological obstacles like the possibility of losing money, despair, substance misuse, limited access to high-quality social support networks, and prejudice [12].

The majority of children living with HIV are entering puberty and having unprotected sex, which puts the population at danger of contracting a resistant variant of the virus. In light of this, it is imperative that significant efforts be made in clinical care, stakeholder support, and research to identify the causes and close the support gap for these vulnerable populations [13]. Adolescents in Ethiopia who are living with HIV/AIDS make up a sizeable section of the population; learning about the lived experiences of this distinct group of people will help ensure that these adolescents have better futures. The majority of the studies on adolescent living with HIV/AIDS that is currently available focuses on the severity of the illness and how ART adherence obstructs knowledge of their lived experiences and the difficulties they have faced. Despite this knowledge, there is still a dearth of literature on the experiences that adolescents with HIV/AIDS in Ethiopia have on the ground [14]. In our nation, there hasn't been a lot of qualitative research done on the lived experiences of adolescents living with HIV/AIDS and taking ART for the whole of their lives.

## Materials and methods

### Study setting and study period

The investigation was carried out in Bahir-Dar city at the Felege Hiwot Comprehensive Specialized Hospital. The capital of the Amhara National Regional State, Bahir Dar, is located 565 kilometers from Ethiopia's capital, Addis Ababa. There are 422,580 people living in the city overall, according to the report from the service officer and municipal administration plan. Of whom, 215,516 (51%) were females. A 13,869 people are receiving ART at sixteen public and private ART center health facilities inside this city administration. Out of them, 536 were teenagers receiving follow-up care for ART. ART is available in Felege Hiwot Comprehensive Specialized Hospital, one of the public health facilities. This hospital was founded in April 1963, under the reign of Emperor Hailesilasie, and is a referral facility. The study project ran from March 25, 2023, to April 30, 2023.

### Study design

Adolescents living with HIV/AIDS were asked to describe their life experiences using a phenomenological method. By examining people's daily experiences, a phenomenological technique is utilized to try to understand people's feelings and characterize the universal core of a phenomenon [15]. The sensitive issue of the adolescents living with HIV is explored by using such like study design.

### Study participant recruitment procedure

Adolescents living with HIV/AIDS who were specifically chosen and receiving ART follow-up at Felege Hiwot Comprehensive Specialized Hospital were enlisted. Included were teenagers who had volunteered for an in-depth interview, were free of comorbidities, and had known their HIV status for at least two years. To recruit study participants, a succinct explanation of the study's purpose was provided to the ART center focal person, medical staff assigned to the

ART clinic, and team leaders of teenagers. After that, health practitioners, team leaders of Operation for Triple Zero (OTZ), and psychosocial program administrators provided a list of teenagers who knew their HIV status more than two years ago and who had attitudes toward open communication about themselves.

To recruit study subjects first, the nurse calls the teenager when they are picking up their prescription, inquiries about their willingness to take part in the study, and then directs them to the data collector. Second, the principal researcher provides thorough information about the research and obtains consent from teenagers and family, in that order. The interview was then held in a quiet, separate room.

## Sample size determination

Information saturation dictated the sample size. Thus, thirteen carefully chosen teenagers on ART were included based on this methodology. The data was saturated after the tenth participant, but three more people were questioned to make sure the saturation threshold had been reached and to record any new information that could have appeared.

## Data collection method and tools

For in-depth interviews, interview instructions were first written in English and then translated into Amharic, the native tongue. The interview guide took into account the socio demographics of the patients as well as the real-life experiences of teenagers living with HIV/AIDS. The information was elicited by semi-structured open-ended interview guiding questions, which included questions to address the study's goal through the application of probing techniques such as how, what, and why inquiries during the data collection period.

## Data collection procedure

To prevent disruptions during the interview, a quiet and distinct room was selected for the interview in the hospital. Throughout the interview, an audio recorder was activated. The principal investigator gathered all of the data while properly investigating each subject in relation to the study's goal. In a calm environment, the interview was done in person. The participants were asked to start the interview at one point. Interviews continued until conceptual saturation was achieved, that is, until no further information could be gleaned. How and why were used in conjunction with the probing technique to obtain sufficient information about the point of interest. Based on their interview sequence, individuals were categorized as participants (P): P1, P2, P3, etc. in accordance with their interview order. Throughout the interview, the principal investigator's reflections, body language, and verbal and nonverbal clues were noted by the interviewer in field notes, and the audio recording was marked with an identification number. After all, the interviewer has answered all of the participants' questions and given them an opportunity to talk about or bring up any other topics they would like to. After the interview was over, the interviewer thanked them one last time and set up a follow-up call to make sure the transcription accurately reflected what the participant had said.

## Trustworthiness

Saturation of data, audio recording, verbatim transcription, member verification (reviewing the data record, transcribed data, and interpretation by the participant who gave the data), and a thorough field note were all used to ensure reliability. To ensure the validity of the results, coauthors and specialists conducted an external inspection using the preliminary data findings. To ensure dependability, an audit trial was conducted with a thorough description of the research

from the start of the study to the writing of the findings report. To validate the quality of the data probed, a rich, textual account of the lived experiences of the adolescents living with HIV was also conducted. Furthermore, during the data collecting and analysis process, suitable probing of the data and a comprehensive explanation of the outcome with quotes were utilized.

## Data management and analysis

The data analysis was made easier with the use of the ATLAS.ti 8 program. Both data collection and analysis were done at the same time. Daily work involved transcribing the data verbatim. The principal investigator and linguists conceptually translated the transcribed material. Six phases for the analysis process were implemented, and the data was analyzed using thematic analysis. The principal investigators read the transcribed data several times to become familiar with it. They then generated the initial codes and arranged them according to how similar they were. They then thoroughly searched for themes, reviewed them, and finally named and defined them for thematic analysis. Inductive generation of codes was used. Irrelevant codes were eliminated and similar codes combined.

Sub-themes, or categories, developed from codes with related concepts. Categories containing comparable concepts were used to construct themes. The results were evaluated in light of the available information. To draw attention to each area and demonstrate connections with each theme, quotes were employed. The final topic analysis had been approved by the lead investigators. The accuracy and consistency of the data were verified by comparing the interpretations of the qualitative data with the verbatim transcripts.

## Ethical consideration

With the ethical approval procedure number CMHS/IRB/01/757/2023, ethical approval was acquired from the Institutional Review Board of the Bahir Dar University College of Medicine and Health Sciences. The Amhara Public Health Institute (APHI) provided the study site with a letter of permission. Teenagers' and parents' written informed consent and assent were acquired, respectively. A comprehensive explanation of the study's purpose, advantages, and hazards was given to each participant. Following an explanation of the study's objectives, each participant was asked if they could capture audio. During the data collection, interview, analysis, and reporting phases, their name was substituted with a code identification number. The core ethical precepts of autonomy, beneficence, non-maleficence, secrecy, and the pursuit of justice were upheld throughout the whole research process (Table 1).

**Table 1. Socio-demographic status of the study participants on the experience of living with HIV among adolescents at Felege Hiwot Comprehensive Specialized Hospital, Bahir-Dar, Northwest Ethiopia, 2023.**

| Participant 01 | F | 18 | Grade 12 |
|---|---|---|---|
| Participant 02 | M | 17 | Grade 1 1 |
| Participant 03 | M | 18 | Grade 9 |
| Participant 04 | M | 18 | Grade 12 |
| Participant 05 | F | 15 | Grade 8 |
| Participant 06 | F | 19 | Grade 12 |
| Participant 07 | F | 18 | Grade 11 |
| Participant 08 | M | 15 | Grade 8 |
| Participant 09 | M | 15 | Grade 8 |
| Participant 10 | F | 16 | Grade 10 |
| Participant 11 | F | 17 | Grade 9 |
| Participant 12 | M | 18 | Grade 10 |
| Participant 13 | M | 19 | Higher educational level |

**Table 2. Themes, sub-themes and codes emerged from the data of the experience of living with HIV among adolescent at FHCSH, Northwest Ethiopia, 2023.**

| Main themes | Sub themes | Codes |
|---|---|---|
| Lived with burden of HIV/AIDS | Psychological stress | Inferiority, silence, lack of interest, unhappy life, worry/thinking more, talking alone, loss of family, missing class/school, stereotyped/seen as crazy |
| | Stigma and discrimination | Isolation, gossips/rumors, talks from other people, negative attitude of community, fear |
| | Negative emotional feelings | Sadness, crying, angry, shocked |
| Disclosure | knowing their own HIV status | Told indirectly, know, disclose |
| | Disclosure of HIV status to others | Hidden, secret, no comfort to talk freely, careful to talk |
| ART adherence | Habit of ART intake | Refuse the treatment, stopping the medicine/missing the dose |
| | Barriers to ART adherence | To many tab, interferes with fasting, concern of religion, fear of taking in front of others |
| | Facilitators for ART adherence | Help, remembering them alarm from peer and family |
| | | Hope, hardship in relationship, |
| Future aspiration | Sexual relationship | no freedom to have a friendship |
| | Forming family/being married | Independency future wish |
| | Demand of support | Financed, lack of support suffering for transportation fee, need |
| | Demand of support | Financed, lack of support suffering for transportation fee, need |

## Emerged themes

Following a thorough analysis of the data, Table 2 below illustrates the four major themes and eleven supporting themes that surfaced. The findings were then synthesized, and each of the emerging main themes and related subthemes was put down. Lastly, sample quotes from the participants' data analysis provided credence to the synthesis of the findings (Table 2).

## Theme 1: Lived with burden of HIV/AIDS

Adolescents with HIV/AIDS have to deal with numerous challenges throughout their lives. The youngster claimed that upon discovering their HIV status, they experienced feelings of impending mortality along with sadness and despair. Throughout their lives, they experienced a number of issues, including psychological difficulties associated with taking daily medications, anxiety brought on by a fear of social stigma and discrimination, fear of illness and early death, and a lack of freedom in school due to discussions and rumors about HIV/AIDS from peers and the community that upset their feelings and make them live horrible lives. Furthermore, the adolescents had concealed their use of ART, and a majority of them continued to not follow the prescribed course of action out of concern about being found out.

**Subtheme 1: Psychological stress.** HIV was described by participants as physically and emotionally painful. Due to HIV/AIDS, they have suffered from disease and lost parents, siblings, and other family members. Their experiences with their parents' and relatives' deaths have demonstrated to be upsetting. Concerns about their impending demise had also caused

them to undergo psychological anguish. Adolescents are aware that their family's faults have caused them to live an unpleasant life, even though they did not cause the condition to develop. As any other person living with a virus, the majority of participants clarified that they are not like other people since they take medication on a regular basis, which has an impact on their mental state. As a result of contracting the virus, each participant had some degree of psychological anguish and emotional vulnerabilities throughout their life. When the respondents consider their HIV status, the majority of them experience depression and general disinterest in everything. HIV bothered them since it impacts their mental health.

As an illustration, an 18-year-old female participant said, *"HIV has its own problems." Since the virus is ingrained in my blood, I accept that I will never be happy and live my life through medication.* (Participant: 07).

Due to psychological disruption brought on by HIV, participants worry more because they fear social isolation and negative comments about HIV from classmates, neighbors, and other people. Their mood is affected by this, which makes them feel hopeless and depressed. Adolescents living with HIV therefore had awful emotions. They have experienced emotional distress as a result of the virus. A 17-year-old female participant's statement, which goes like this: *"I have felt a terrible sense. . . I suffered emotional harm.* (Participant: 01).

Some participants describe living with HIV/AIDS as nothing more than taking medication; they consider themselves to be on par with other people and endure great suffering as a result of their daily medication regimen and underlying medical condition. A male participant, eighteen years old, endorsed this viewpoint, saying, "*No one likes disease. . ., but even if the disease comes, I am living by taking medicine with some stress."* Participant number: 03.

Living with HIV is linked to taking antiretroviral therapy (ART) for the rest of one's life, which makes the patient sad. Teenagers have experienced emotional upheavals that have made them hopeless about everything. Furthermore, they experienced emotional disturbances such as anxiety, worry over the virus that killed their family members, fear of dying young, and despair over the course of their lives. An 18-year-old female participant, for instance, stated: *"I can get sick or bedridden if I don't take the medication as prescribed. This prompts me to reflect more. Since this illness is fatal, I shall take my medication as directed. (*Participant: 01.)

Teens living with HIV had a difficult time since discussions about the virus were uncomfortable and social media ideas upset their feelings.

**Subtheme 2: Stigma and discrimination.** The majority of the participants experienced anxiety as a result of being afraid of gossip, feeling alone and stigmatized, and facing prejudice from their peers, neighbors, and the society at large.

Adolescent reports indicate that because stigma against HIV-positive individuals persists, the community has a negative attitude toward the diagnosis of HIV/AIDS.

All of the individuals had experienced discrimination and isolation from the community; however the majority had not encountered any stigma or discrimination from their guardians, family, or close friends. A participant, tears in their eyes, speaks very heartbrokenly about their experience with discrimination and stigma. An 18-year-old female participant, for instance, provided the following explanation: *"She has HIV! Why are you playing together?. . . That was the only time I felt threatened as a child and there was every occasion where I was told that I was not looking at the store to buy something for myself. There is a big problem from the community. What can I tell you; it is very difficult! There is a lot left. . ."* (Participant: 07).

The community's negative attitude toward HIV/AIDS and the fact that teenagers living with the disease are unable to openly connect with society are two things that have not altered. As a result, the teenagers encountered many difficulties from their peers, instructors, and even the neighborhood. Participants clarify that people who are aware of their HIV status avoid playing with them out of fear for HIV, talk about HIV more in front of teenagers, and

withdraw from their group or team, all of which cause psychological issues and negative emotions in teenagers. An eighteen-year-old female participant, for instance, clarified as follows: *As an example, my friend just told me that if I had HIV, I would commit myself. This gives kids the chance to think of someone with HIV as less than human. There has still been discrimination and isolation from the community. (*A participant: 07).

Many individuals kept their HIV status a secret out of fear of prejudice and stigma from the community. An 18-year-old female participant echoes this sentiment when she says, *"I do not tell my HIV status for others because there is still stigma and discrimination."* (Participant: 01).

Teenagers are not comforted by people's conversations and rumors concerning the virus. Negative explanations of HIV are more common, and the focus is mostly on societal stigmatization of the adolescent. *"I don't want to talk to them because now the society is said to be educated, but in practice it remains, the interviews coming from them are not good,"* says a male participant, 17 year-old, in support of this opinion. (Participant: 02).

HIV was not well-regarded in the community, and those who had it are stigmatized. An 18-year-old female participant explains, *"One of them even said that my mother stays away from a person with HIV,"* which lends credence to this. (A participant, 07*)*.

Teens said that they struggle with unwarranted conversations about their HIV status from peers, gossip, or hearsay, and that these conversations cause the child to be stigmatized at school. They are unable to play with them freely as a result, and they plan to boycott school. Because they fear prejudice and stigma after announcing their status, the majority of participants avoid taking ART medication in public. An 18-year-old female participant, for example, clarified as follows:

*"I will bring water into my room and take it, or I will take it outside, if a guest unexpectedly shows up. I take the necessary action sooner because discrimination and stigma still exist"* Participant: 01).

**Sub theme 3: Negative emotional feelings.**   The participants expressed sentiments of melancholy, dread, and anxiety over their illness and viewed HIV as a negative aspect of their lives. When the participants learned their status for the first time, the majorities of them were astonished and sobbed a lot. In addition to this experiencing intense fear, anger, silence, and loneliness were among the emotional responses that the majority of participants reported. A female participant, 19, provided the following argument to bolster this:

*"It was challenging for me since I didn't adjust to the medication, I had trouble falling asleep, and I would experience psychological issues."*(Participant: 04).

Some of the participants were informed early in childhood and were not confused about why they were taking medicines, but the majority of them were confused about the purpose of taking their prescription and attempted to find the true information throughout their lives. Some of the participants were informed that the medication was being taken for another illness, and they were unaware that they were taking it to treat the virus. Most participants use their effort to gauge who they are. One participant, a female who was 19 years old, stated that:

"*This depressed me for a long time, and I used to cry every day. One day, my father noticed me crying and asked, "Why are you crying?" To my father, I asked, "Why am I taking this medication?" My friend finds it offensive."* (Participant: 06).

## Theme 2: Disclosure

Adolescents who are HIV positive experience various impacts on their lives as a result of HIV disclosure. Some of the consequences observed after disclosure were psychological anguish and emotional disturbance. Teenagers found it challenging to openly disclose their HIV status to others due to their fear of stigma and other people's stigmatization. A majority of the study participants expressed their anxiety that others will learn of their HIV status and disseminate the information, leading them to prefer keeping their status private and hidden from others.

**Subtheme 1: Knowing their own HIV status.**   The subjects learned about their HIV status in early adolescence, despite having HIV from birth. Teenagers go through a difficult and crucial stage in their lives when they find out their HIV status. Reactions included feelings of melancholy, rage, pessimism, impending death, and anxiety about the implications of their HIV diagnosis for their future. One 19-year-old female participant, for example, stated that:

"*I cry and laugh, and my happiness is coming to an end." Nevertheless, at the time it made me cry.*"

(Participant: 06).

The participants were perplexed about taking medication on a regular basis and attempted to find out the truth regarding drug use throughout their lives; however, a few were informed at an early age and were spared this difficulty. A few interviewees stated that their parents had given them an explanation about their status. A participant who is 17 years old supports this notion as: "*I was eleven years old when my father told me.*" (Participant: 11). Through their own efforts, the participants discovered that they were infected with the virus, and they independently ascertained the rationale behind their daily medication use. For instance, a participant who was eighteen years old stated that:

"*No one informed me that everything was here when I arrived, and there is a poster." I read, I observe, and I always remember the lesson. Here's how I discovered it.*" (Participant: 07).

Peer to peer discussions in psychosocial settings and the Operation Triple Zero program assist individuals in self-discovering their HIV status and helping them come to terms with the fact that they are infected. Indirect information regarding the virus was disclosed during this conversation by program team leaders and medical experts. They discover that the medication must be taken every day for the rest of their lives. Certain participants claimed that the medication they were taking was for a different illness, and they were unaware that it was being used to treat a virus. For instance, a participant who was eighteen years old stated that:

"*The family was telling me that the medication was for a different illness, but I didn't know any of that at the time. The nurse responded to my question in a straightforward and suitable manner, and the association has enrolled me here.*" (Participant: 03).

**Subtheme 2: Disclosure of HIV status to others.**   Participants are afraid of prejudice and stigma, thus they don't want to reveal their status. With the exception of their close friends, family, and guardians, nearly all of the participants concealed their HIV status from outsiders. Teenagers who hide their status from others out of fear of loneliness, pointless conversations, and rumors in the neighborhood experience psychological pain and low self-esteem. Participants take ART medication cautiously, and they have never taken it liberally in public out of concern that their status will be inadvertently revealed. Participants keep their status a secret

from other people. They are experiencing psychological disruption as a result of their everyday medication usage and anxiety related to disclosure issues. For instance, a male participant who was 17 years old stated that: "*I don't want to speak with them since, despite claims to the contrary, society is still uneducated today. Their conversations aren't very excellent.*" (Participant: 02).

The study's participants were cautious about sharing their status with others and, in the event that they did, they were cautious about whom they should confide such private information. According to the participants, they tried to live their lives in secrecy from other people. After disclosing to their families, the participants stop worrying about them. Subsets of individuals choose to reveal their HIV status only to close friends.

One participant, a male who was 19 years old, stated that:

*"Although I kept it a secret, one of my pals is aware of my position, "I believe my classmates will talk about me if they find out."* Participant: 06).

## Theme 3: ART adherence

Every participant had doubted the need for medication throughout their lives and had questioned what illness their family members and themselves were taking it for. This inquiry was made prior to their serostatus being known, and even after learning their status, they continued to not follow the ART guidelines. However, they stick to the ART program once they feel emotionally stable and reassured about themselves.

**Subtheme 1: Habit of ART intake.** Some teenagers purposefully refuse to take their drugs, and parents find it difficult to persuade them because of the questions they pose, such as "Why am I the only one who takes medicine every day?" What went wrong with me? They consequently stop taking their medications, which results in poor ART adherence and a lower length of stay in care before patients realize they are HIV positive. After learning all there is to know about HIV/AIDS, adolescents continue to take the medication consistently and stick to their treatment plan. The hospital's Operation Triple Zero (OTZ) and psychosocial programs play a significant part in encouraging patients to stick to their ART regimen. A 17-year-old female participant, for instance, clarified that:

*"I used to take medication when I was pleased and stop when I was upset, but since I started the OTZ program, I have made the necessary corrections. Since 2022, I have been taking medication on a regular basis".* (Participant: 11).

In addition, some teenagers refuse to take medication after learning of their serostatus because they believe they have nothing to do with their HIV infection and instead bring up the matter with their family. For instance, a male participant who was 19 years old stated that;

*"I stopped taking the medication after learning that I have HIV. I was unable to listen to my father's advice to take it, therefore I stopped taking medicine for six months." (*Participant: 13). Instead of continuing with the medication, they would rather turn to their faith for healing and believe that God would enable them to fully recover.

**Subtheme 2: Barriers to ART adherence.** The individuals found taking ART difficult because of daily drug schedules, side effects, pill burden, and problems with fasting. Participants stated that they had suffered from ART medication side effects and pill burden.

One participant, a boy who was eighteen, stated, "*I was thinking about the amount of medicine I took.*" (Participant: 04). Adolescents who are confused about taking medication tend not to follow through on their treatment. Furthermore, the fact that they contracted the virus without their consent causes them to refuse treatment, and the majority of adolescents have stopped taking medication as a result of emotional distress during the course of the illness. A few of the participants had engaged in religious healing, receiving healing from a God in whom they had faith, while also taking their medications and disobeying family and medical professionals' advice. Due to conflicts between medication intake times and fasting, taking medication in the morning violates fasting. Some participants refrain from taking medication on a regular basis.

One participant, a female who was eighteen, stated, "*I might have to eat throughout my fast. Why don't I fast when someone else is? I go through a lot.*" (Participant: 07). The participants shared that they had forgotten their medication several times when they were out and about and had thus missed their dose. They left their medication behind and headed somewhere else. An eighteen-year-old female participant lends confirmation to this, saying,

> "*I went somewhere and when I left, I forgot the medicine. I gave up taking medication.*" (Participant: 01).

### Subtheme 3: Facilitators for ART adherence

Almost all individuals living with HIV/AIDS reported that maintaining adherence to treatment is greatly influenced by the support of a caregiver and the OTZ program. Some of the participants used a smartphone alarm to remind them to take their medication on time and consistently, never missing a dosage. They are reminded to take the medication by their close friends and relatives. Adolescents learned the value of treatment adherence through peer-to-peer conversations and the hospital's OTZ program. Peers, family, and health professionals alert the teenagers to take their medication on time in all aspects, which helps them, stay compliant with the ART therapy, in addition to counseling and advice. One 19-year-old female participant gave an example of this as follows:

> "*My neighbor also called to remind me to take the medication, and my colleague warned me to hurry because it's now too late. These enable me to take my medication on schedule.*" (Participant: 06).

Support from family and or guardians gives adolescents hope for the future and greatly helps them stick to their routine. For example, a 15-year-old female participant said that: "*My families do everything I ask because they think they can hear what they are thinking. Any of my brothers with the virus in their blood will do anything I ask.*" (Participant: 05). The participant stated that they are happy with the attention they received from the ART clinic staff and that the health services offered in the clinic for HIV/AIDS are fantastic. A female participant who is 19 years old supports this notion as follows:

> "*They treat us like sisters, brothers, and moms in their line of work, and the service they provide is really great*". (Participant: 06).

The majority of medical personnel treat patients well, and they are frequently thanked by their patients. For instance, a participant who was eighteen years old stated that:

*"It wouldn't be an overstatement to say, in all honesty, that we were raised by professionals rather than our family. The professional attention that is provided is wonderful."* (Participant: 03).

## Theme 4: Future aspirations

Most of the participants have hope and dreams for what they want to be in the future life. Participants wish to pursue their lives independently of other people's support and have no desire to live with their relatives. Most of the participants hope to marry a supportive partner and start a family in the future.

**Subtheme 1: Sexual relationship.**    The majority of participants had plans to have a boyfriend or girlfriend, but none of them were in romantic relationships or had ever engaged in sexual activity. Participants favor similar infected individuals for their upcoming plans. A few participants stated that they presently have no desire to have a sexual relationship before completing the course and those they intend to start one after completing their education and earning money.

This is supported by a 18- year- old male participant as: *"I'm not in a romantic relationship. When I grow up, that's what I always think."* (Participant: 04).

The participants declare that they have no plans to date anyone sexually, either now or in the future. A 15-year-old female participant gave an illustration of this when she said, "*I can't tell you this because God only knows about the future and we don't know the terms of tomorrow." I'm not concerned about it."*(Participant: 05). The participant, who was eighteen years old, provided an answer to the question, "Do you have a boyfriend/lover?" stated as "*No, I'm studying right now, so I don't think about it."* (Participant: 07). In their future lives, some participants hope to have a spouse with a matching serostatus.

## Subtheme 2: Forming family/being married

The majority of interviewees stated that they hope to marry an HIV-positive person in the future. This would facilitate their comprehension of one another throughout their lives. In the future, participants want to start a family and raise kids. The majority of them want to start a family after being financially secure and self-sufficient. They state that their family needed to marry their teenager to the same person since they wanted a partner in the future.

A male participant who was eighteen years old, for instance, stated: "*I don't have any sexual relationships other than friendships from class or the neighborhood, I always believe that when I grow up. . . I'll have to bring up a family."* (Participant: 04). A few individuals expressed concern about getting married to their desired partner due to their HIV status. A female participant of eighteen years gave this example: "*The future's creator knows, yet even if there's a terrible person, I can't manage it. It's challenging. I would rather it be someone similar to myself, even if I have it."* (Participant: 01)

Some individuals said they didn't give a damn about their spouse's HIV status. We can continue dating an HIV-negative person if the other person is at ease with it and agrees that he would marry an HIV-positive person provided that HIV transmission can be stopped.

One male participant, 17, gave the following example: "*I don't think there is any problem if she accepts me." We use condoms when we have sex if she accepts me."*(Participant: 02).

**Subtheme 3: Demand of support.**    Participants stated that both their families/guardians and the medical staff at hospital ART clinics are providing them with excellent assistance. The majority of participants expressed their strong need for employment possibilities so they may

live independently and their strong desire for support from the government, stakeholders, and community from various angles.

Furthermore, nearly all of them had aimed to obtain more effective medication, administered once or twice a year, to treat HIV/AIDS in the future. Teens are supported appropriately by the family members of participants and their guardians, who are always there for them no matter what. According to several participants, one coping mechanism that helps them while living with HIV/AIDS is family support. This helps them to go on with their lives and not give up. Due to the continued stigma and prejudice against the HIV/AIDS community, the majority of participants stated that there is a misunderstanding of the disease in society.

The participants also said that it would be preferable if HIV positive people were not discriminated against. The participant seeks support from society from a variety of angles, and society ought to place a strong focus on their opinions regarding the rejection and exclusion of those who are infected with the virus. Teenagers had experienced difficulties with social exclusion. Some people don't think HIV-positive individuals are human, and they don't treat them with dignity. Every participant expressed a strong need for financial, material, and nutritional support as well as the creation of job opportunities so they could sustain themselves. The participant asks each pertinent party for a variety of resources.

For instance, a male participant who was eighteen years old stated: *"It would be beneficial if the appropriate body was contacted and offered to support us with food and hygiene supplies. If they can help as much as they can, that's good."* (Participant: 12). According to a different participant, in order for teenagers to remain autonomous, the relevant authorities should provide them with the chance to work and earn money. One participant, a female participant of 19 years old, gave the following example: *"It would be great if there were different job opportunities for the youth."...It would also be fantastic if there was something to encourage a business or organization so that the association could function independently, provided that it was funded."*(Participant: 06).

### Relationship between emerged themes with the experience of living with HIV among adolescents

The relationships between the emerging themes and sub-themes depict the lived experiences of adolescents living with HIV/AIDS. As HIV/AIDS is a disease that can cause physical, psychological, curtains future life and social problems for those who are infected, this fits in well with the emerging theme of "lived with the burden of HIV/AIDS, ART Adherence, disclosure, and future aspiration," which describes how adolescents have suffered from a variety of issues throughout their lives, including daily medication intake, stigma, discrimination, and psychological distress brought on by the disease.

The majorities of teenagers were dissatisfied and concealed their position out of fear of social shame, which led them to follow the regimen irregularly. But social and familial support is crucial for their commitment. Their goals for the future are obscured by the disease's impact, lack of support, and stigma, all of which have plagued them throughout their lives (Fig 1).

### Discussion

This study set out to investigate the lived experiences of young people living with HIV. The goals of teenagers living with HIV, ART adherence, psychological and emotional difficulties, and disclosure issues are all significant global public health concerns. The study's findings highlight four key themes that represent the lived realities of young people living with HIV.

Eleven sub-themes were identified to organize these lived experiences under four main themes: living with the burden of HIV/AIDS, disclosure, ART adherence, and future

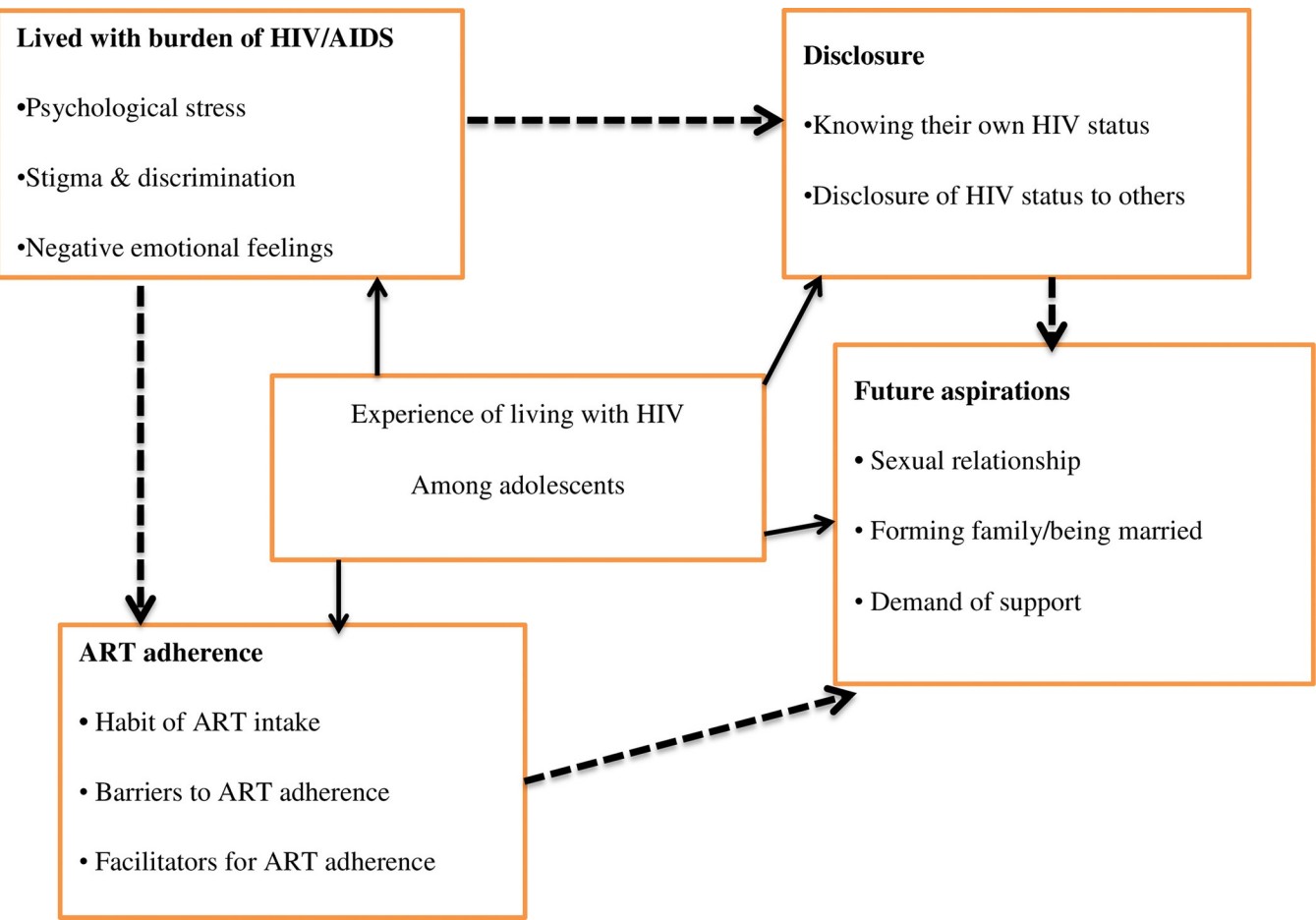

**Fig 1. Emerged themes, relationship of themes and sub themes with the experience of living with HIV among adolescents at Felege Hiwot Comprehensive Specialized Hospital, Bahir-Dar, Northwest Ethiopia, 2023.**

aspirations. Adolescents who were HIV/AIDS sufferers had to bear numerous burdens throughout their lives. Throughout their lives, they have faced a number of issues, including difficulties with daily medicine ingestion, psychological stress, and the harmful impacts of emotions on social relationships. Furthermore, teenagers had missed dosages, which resulted in their non-adherence to the ART regimen. One of the problems teenagers living with HIV/AIDS confront as a result of the stigma and stigmatization by the community is fear of disclosing their status to their friends and neighbors. The participants hope and aspire to live independently; they have no desire to live with the family.

According to the participants, they want to marry someone who is HIV positive in order to start a family. In the future, they hope to wed an HIV-positive partner. The teenage HIV positive person concealed and did not disclose their status to others. This result is consistent with studies conducted in Ghana and Brazil where participants kept their HIV status a secret from everyone outside their family [16, 17]. The study's participants had been nervous about telling people about their status, and even when they did, they were cautious about whom they trusted with such private information. This result is consistent with a study of teenagers living with HIV that was carried out in Ghana [17].

Individuals felt a variety of unfavorable emotions toward HIV. They saw the illness as a problem and something that would have a detrimental impact on their life. These sentiments

included melancholy, anxiety, loneliness, and a sense that they were different from other people.

These results are in line with those obtained in Brazil, South Africa, Tanzania, and Zimbabwe, where adolescents' negative moods outweigh their positive feelings [15, 18–20].

According to some participants, they are able to lead normal lives. Despite the fact that they must look after themselves in order to maintain their health participants think they are capable of accomplishing whatever that a healthy person can. Nothing prevents them from pursuing their desires. This result is consistent with findings from South Africa and Brazil [16, 18].

For a variety of reasons, some participants stopped taking their medications, and they frequently ask the families why only they took the drug and not the others. Following disclosure, the participant no longer asked themselves "why did I take the medicine?" Disclosure also improved their understanding of their illness and led to higher adherence. This result is in line with the Brazilian research [18].

Adolescents can improve their adherence by telling the truth, which has significant clinical implications for them. The majority of individuals learned about their status for the first time via friends, family, doctors, and other caregivers; others learned about it from posters, commercials for ART medications, and how other people handled it on their own. This conclusion is consistent with research conducted in Kenya, where in certain participants ascertain their status through self-incrimination [20].

A few individuals learned about the sickness when their parents discussed it with the medical staff. This result is in line with studies conducted in Swaziland and South Africa [16, 21].

According to the results, participants said that finding out they had HIV caused them to feel hopeless and depressed. They also said that it made them feel like they would die soon, lost interest in their family in all situations, and gave up on everything. This result is consistent with research from Tanzania, Zimbabwe, and Ghana [15, 17, 20].

This research has a practical benefit in that it alerts medical professionals to the need of treating teenagers with similar issues as soon as possible. When the participants initially learned about their status, they saw shock, sobbing, anxiety, anger, and suicidal thoughts in the individuals. In addition, they struggled with their family, were afraid of dying, and reduced their communication with others, all of which are consistent with findings from a Swaziland research [22].

Adolescents learned to deal with their illness as a clinical phenomenon, but they are afraid of other people's opinions, so they keep their HIV status a secret. The primary reasons they were kept a secret from others were discrimination and stigma. People have responded in a variety of ways, including unwarranted conversations and gossip, putting them in the spotlight, and removing them from the situation. These results are in line with studies conducted in Swaziland and South Africa [16, 22].

However, they don't conceal their condition from individuals who are afflicted because mutual understanding would exist. Those who disclosed their status to others received both positive and negative responses. One positive experience was witnessing a friendship grow stronger as a result of feedback that some individuals received care and accepted their illness as normal. Even so, their negative experiences in life had been ending the partnership and separating.

This outcome was consistent with the research conducted in South Africa [16]. The majority of participants believed they would have a positive spouse in the future, therefore they felt it was important to support and understand one another and not spread the virus to others. None of the participants had engaged in sexual activity up to this point, which is important in preventing HIV transmission.

Additionally, they felt that it was necessary to disclose their HIV status to their potential spouse prior to marriage. Contrary to research conducted in Tanzania and Uganda, where the majority of participants preferred to have an HIV + spouse, few participants desired to marry an HIV negative spouse in the future [15, 22].

The participants' views on potential spouses who are similar to them and their practice of abstinence before marriage prevent the virus from spreading throughout the community, which has a significant clinical benefit in managing the condition and controlling viral spread. On the other hand, health professionals are alarmed by teenagers' belief that they will marry unsuitable people in the future and are placing a lot of emphasis on this problem. The ART clinics offered the participants pleasant services, and the medical professionals treated them like family and gave excellent care.

This result is consistent with studies conducted in Zambia and Brazil [18, 20]. They benefit greatly from the ART clinic's psychological services, Triple Zero operation, and tea and coffee program over the course of their lives. They stated that it inspires and supports them in a variety of ways, with remarkable results. They no longer felt like they were the only ones with the illness thanks to triple zero operation program, and they also learned about ART adherence, sexual interactions, and coping techniques for stigmatization. They also feel more comfortable talking openly about their plan. Possessing friends and possibly a future spouse also helps. This result is in line with studies conducted in South Africa and Swaziland [16, 22].

The participants stated that they had been stigmatized by society as a result of having HIV, and they recommended that education be given to society on a constant basis. The two types of support that enable people to get through life are psychological and familial. This result is consistent with research from South Africa [16]. The government's support was deemed satisfactory by the participants; however, they anticipate greater support in the future when scientific advancements lead to the development of more advanced medications or a cure for HIV. They would prefer financial support from the government as well. This participant requirement is comparable to research conducted in Uganda [22].

According to a study conducted in South Africa [16], the majority of participants desired to start a family and have children in the future. However, some of them expressed a desire to forgo marriage and starting a family because they believed their sickness would interfere with their ability to have a sexual life. The participants conveyed a dread of passing away too soon because they feared they would not live up to their dreams before death claimed them, which would lead to psychological issues. This result is consistent with a Ghanaian research [17].

Overall, the study's findings point to a number of benefits, chiefly for medical professionals who were initially trained to recognize the issues facing young people living with HIV and to address them before they become serious. The primary clinical implications of early management of psychological problems, stigma and discrimination, identification of ART intake habits, barriers and facilitators for ART adherence, handling of disclosure issues, and future aspirations are to control and limit the spread of viral infections within the community.

## Limitations of the study

The study just examines the perspectives of the adolescents; it ignores the opinions of family members, guardians, medical professionals, and important informants regarding the lived experience of adolescents living with HIV.

## Conclusion

Thirteen teenagers with HIV/AIDS who are receiving ART follow-up at Felege Hiwot Comprehensive Specialized Hospital had their lived experiences examined in this study. Based on

the four primary themes of "lived with the burden of HIV/AIDS, disclosure, ART adherence, and future aspirations" with eleven perspective sub-themes, the analysis conclusion of this study was determined. Teens who were HIV positive had to overcome several social, psychological, and emotional obstacles.

HIV disclosure and its aftermath affect people living with HIV emotionally disrupt them and force them to deal with hardships throughout their lives. Adolescents found it difficult to openly reveal their HIV status to others and to take medication in public, which has an adverse effect on their adherence to antiretroviral therapy (ART). This fear of stigma and stigmatization by others also made it difficult for them to take their medication. Most of the participants are optimistic about their future and wish to start a family in the future. The majority of interviewees expressed their strong desire for government assistance in obtaining employment so they can live independent lives.

It is recommended that all relevant bodies address the HIV/AIDS epidemic by offering many forms of support to adolescents, facilitating their disclosure, and improving their adherence to antiretroviral therapy. In addition, there should be an ongoing, intense focus on community health education initiatives to raise awareness and alter societal perceptions regarding the stigma and prejudice against people living with HIV/AIDS.

## Recommendations

To address the issues and health requirements of ALHIV, several interventions at the individual, community, and organizational levels are advised to the relevant bodies.

### For government and stakeholders

It is preferable to provide employment possibilities for individuals living with HIV so they may support themselves and live independently, potentially reducing the dual burden of being an adolescent HIV positive. It is preferable to give the community technical assistance and instruction in stress management and illness prevention, with a focus on the important demographic of adolescents infected with HIV.

### For health professionals working at ART clinic

To keep them compliant with the ART regimen and to alleviate them of psychological discomfort, it is preferable to support the counseling service that focuses on coping mechanisms for psychological problems.

It is preferable to provide the community with ongoing, comprehensive health education about stigma and discrimination.

### For researcher

Further study should incorporate the families, guardians, health care providers, and key informants to triangulate the data on the lived experiences of adolescents living with HIV/AIDS.

## Supporting information

**S1 Annex. Semi structured open ended interview guiding questions in English.**
(DOCX)

**S1 File. Transcribed data.**
(DOCX)

**S2 File. Translated data.**
(DOCX)

## Acknowledgments

It is our pleasure to say thanks to Bahir Dar University for ethical approval and giving us ethical clearance to conduct this study. We are also indebted to thank Amhara Public Health Institute for offering us support letter to pursue the study. Last but not the least we would like to extend our gratitude for study participants and staffs of Felege Hiwot Comprehensive Specialized Hospital for their support during the data collection process.

## Author Contributions

**Conceptualization:** Bazezew Asfaw Guadie, Minyichil Birhanu Belete, Meseret Mekuriaw Beyene, Ayalew Kassie Melese, Tarik Yenew Tessema, Yinager Workineh.

**Data curation:** Bazezew Asfaw Guadie, Minyichil Birhanu Belete, Meseret Mekuriaw Beyene, Ayalew Kassie Melese, Tarik Yenew Tessema, Yinager Workineh.

**Formal analysis:** Bazezew Asfaw Guadie, Minyichil Birhanu Belete, Meseret Mekuriaw Beyene, Ayalew Kassie Melese, Tarik Yenew Tessema, Yinager Workineh.

**Funding acquisition:** Bazezew Asfaw Guadie, Minyichil Birhanu Belete, Meseret Mekuriaw Beyene, Ayalew Kassie Melese, Tarik Yenew Tessema, Yinager Workineh.

**Investigation:** Bazezew Asfaw Guadie, Minyichil Birhanu Belete, Meseret Mekuriaw Beyene, Ayalew Kassie Melese, Tarik Yenew Tessema, Yinager Workineh.

**Methodology:** Bazezew Asfaw Guadie, Minyichil Birhanu Belete, Meseret Mekuriaw Beyene, Ayalew Kassie Melese, Tarik Yenew Tessema, Yinager Workineh.

**Project administration:** Bazezew Asfaw Guadie, Minyichil Birhanu Belete, Meseret Mekuriaw Beyene, Ayalew Kassie Melese, Tarik Yenew Tessema, Yinager Workineh.

**Resources:** Bazezew Asfaw Guadie, Minyichil Birhanu Belete, Meseret Mekuriaw Beyene, Ayalew Kassie Melese, Tarik Yenew Tessema, Yinager Workineh.

**Software:** Bazezew Asfaw Guadie, Minyichil Birhanu Belete, Meseret Mekuriaw Beyene, Ayalew Kassie Melese, Tarik Yenew Tessema, Yinager Workineh.

**Supervision:** Bazezew Asfaw Guadie, Yinager Workineh.

**Validation:** Bazezew Asfaw Guadie, Minyichil Birhanu Belete, Meseret Mekuriaw Beyene, Ayalew Kassie Melese, Tarik Yenew Tessema, Yinager Workineh.

**Visualization:** Bazezew Asfaw Guadie, Minyichil Birhanu Belete, Meseret Mekuriaw Beyene, Ayalew Kassie Melese, Tarik Yenew Tessema, Yinager Workineh.

**Writing – original draft:** Bazezew Asfaw Guadie, Minyichil Birhanu Belete, Yinager Workineh.

**Writing – review & editing:** Bazezew Asfaw Guadie, Minyichil Birhanu Belete, Meseret Mekuriaw Beyene, Ayalew Kassie Melese, Tarik Yenew Tessema, Yinager Workineh.

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
