## [Decision Letter · Decision Letter 0]

8 Apr 2024

PONE-D-24-01002The Experience of Living with Human Immunodeficiency Virus among Adolescents, A Phenomenological StudyPLOS ONE

Dear Dr. Guadie,

Thank you for submitting your manuscript to PLOS ONE. After careful consideration, we feel that it has merit but does not fully meet PLOS ONE’s publication criteria as it currently stands. Therefore, we invite you to submit a revised version of the manuscript that addresses the points raised during the review process.  **Address all comments made by the two reviewers to improve your manuscript.** Please submit your revised manuscript by May 23 2024 11:59PM. If you will need more time than this to complete your revisions, please reply to this message or contact the journal office at plosone@plos.org. Please include the following items when submitting your revised manuscript:A rebuttal letter that responds to each point raised by the academic editor and reviewer(s). You should upload this letter as a separate file labeled 'Response to Reviewers'.A marked-up copy of your manuscript that highlights changes made to the original version. You should upload this as a separate file labeled 'Revised Manuscript with Track Changes'.An unmarked version of your revised paper without tracked changes. You should upload this as a separate file labeled 'Manuscript'.If applicable, we recommend that you deposit your laboratory protocols in protocols.io to enhance the reproducibility of your results. Protocols.io assigns your protocol its own identifier (DOI) so that it can be cited independently in the future. For instructions see: https://journals.plos.org/plosone/s/submission-guidelines#loc-laboratory-protocols. Additionally, PLOS ONE offers an option for publishing peer-reviewed Lab Protocol articles, which describe protocols hosted on protocols.io. Read more information on sharing protocols at https://plos.org/protocols?utm_medium=editorial-email&utm_source=authorletters&utm_campaign=protocols.

We look forward to receiving your revised manuscript.

Kind regards,

Sogo France Matlala, PhD

Academic Editor

PLOS ONE

Journal Requirements:

2. In the online submission form, you indicated that most of the data are included in manuscript but some of the data like audio records, transcription and translations are restricted for the sake of ethical issue. The supporting information and other data will be available via corrospondence email adress.

Reviewers' comments:

Reviewer's Responses to Questions

**Comments to the Author**

1. Is the manuscript technically sound, and do the data support the conclusions?

Reviewer #1: Yes

Reviewer #2: Partly

2. Has the statistical analysis been performed appropriately and rigorously? 

Reviewer #1: Yes

Reviewer #2: N/A

3. Have the authors made all data underlying the findings in their manuscript fully available?

Reviewer #1: Yes

Reviewer #2: Yes

4. Is the manuscript presented in an intelligible fashion and written in standard English?

Reviewer #1: Yes

Reviewer #2: No

5. Review Comments to the Author

**Reviewer #1:** Reviewer comments

Overall, the paper is well written, and theoretically informed. Saying that, here are some of my comment and suggestion for the better improvement of the manuscript.

1. Page 7:

Abstract: Line 17-18: The authors should better consider Re-placing words; For example, in line 17-18 and I quote “But the lived experiences of adolescents living with the 18 human immunodeficiency viruses were not well investigated before in Ethiopia.” The “But” could be replaced by ‘’To the authors knowledge …...” as there could be published or unpublished research paper work undertaken in that specified area.

2. Page 8:

Line 39: The authors should better to clarify here in the Abstract part (line 39) or Line 696 of the conclusion part; what a “Persistent health education” is and what the content of it to make it understandable for an international readership.

3. Page 10

Line 74: The authors could better replace “Study area” with its qualitative equivalence “Study setting”.

4. Page 11

Line 91-93: The reader needs to be aware of why the selected study design (phenomenological approach) is appropriate/fit to answer the objectives of the research question rather than generally describing the general truth of the design.

5. Page 11

Line 95: The authors state that they recruited a purposive sample of patients for the interviews. They need clearly describe which type of purposive sampling techniques used/chosen and its appropriateness for the study.

6. Page 11

Line 99-100: The authors should better to describe who are “team leaders of adolescents” and how they were appropriate for recruiting study participants.

7. Page 11

Line 100-102: “……the list of adolescents who knew their HIV status more than two years ago and had attitudes towards free talk about themselves……” The authors should better to describe the reasons of the inclusion cutoff point >2years for recruiting participants who knew their HIV status more than two years is extremely concerning to me as the research studies these days should be inclusive of diverse type of patients to get a good picture of their lived experience.

8. Page 13

Line 136-137: Inconsistency with result quotation; As there is no any result quotation using a body language or nonverbal cues, so please either provide at least one quote or edit it.

9. Page 15-16

Line 186-201: The authors needs to elaborate more on the analysis. The authors indicate that they used the ATLAS.ti 8 software. While this is certainly laudable, it is important to know how they practically apply the software from the initial steps to how their main themes were generated, what they were and how they assured robust practices in analysis.

10. Page 17-19

It would be useful to know more about the process of selecting the four main themes in relation to lived experience of ALHIV. Was consideration given to additional/alternative aspects, and why were these rejected for example; the quality of care given by the health facilities & the care given by HCP, availability & accessibility of medicine, etc...

11. Page 18-19

Table 2: “The emerged data main themes seem valid, but the emerged sub-themes and codes used to illustrate that don’t seem/mixed to fit entirely under each theme”. For example, the authors should better to describe why the Sub theme

“Frustration to disclose their condition to others” is not part of the “Stigma and discrimination” or not.

The authors should better to describe why the Sub theme “Habits of medicine intake” is not part of the “Barriers to ART adherence” or not.

The authors should better to describe why the main themes “Future aspiration” is the major identified Theme as the objective of study is mainly on the lived experience of ALHIV.

12. Result; “While the study appears to be sound, the language especially the quote is unclear, making it difficult to follow. I advise the authors work with a writing coach or copyeditor to improve the flow and readability of the text.”

13. Result; Results are quote heavy be more selective about use of examples, i.e., do not provide an example to every theme/sub-theme.

14. Page 34

Line 557-570: Rather than putting this in writing here, the information would be better moving and merged with line 218-221 for easy understanding, flow and to enhance readability.

15. Page 35

Discussion: I would suggest the discussion section is too many and better to make more specific on the major identified reasons and make it short and clear for easy understanding and enhance readability without losing the empirical grounding and relevant context.

16. Page 40:

Line 696: The authors should better to clarify in the Abstract part (line 39) or here Line 696 of the conclusion part; what a “Persistent health education” is and what the content of it to make it understandable for an international readership.

17. Page 49:

Figure 1 gives a good description of your results, but it need to show a clear clarification on how one main-theme is related to another main-theme and even how each sub-theme in the main theme is related other sub-themes.

**Reviewer #2:** This is an interesting piece of work and I would like to commend the authors for the effort made in data collection, however, This manuscript present a significant concept and they have researched it adequately, however i have noted the following:

The introduction does not address the topic- it is generalized to the HIV positive community, adolescents are only mentioned as an example not the main focus and also does not provide the overview of the problem- introduction should quantify the problem, globally and locally. Authors have presented their results well but it lacks technicality and scientific writing element.

The main aim of research is to make practical recommendations, this manuscript lacks that.

In addition to the comments, I would recommend that the manuscript undergoes copyediting to improve the technicality and grammatical errors

6. PLOS authors have the option to publish the peer review history of their article (what does this mean?). If published, this will include your full peer review and any attached files.

Reviewer #1: **Yes: **Habtamu Wondiye Bekele

Reviewer #2: **Yes: **Rirhandzu Mabasa

---

## [Author Response · Author response to Decision Letter 0]

27 Jun 2024

we would like to say Thanks about your constructive feedbacks and comments given for our manuscript.

here we are ready to hear from you about coming comments and suggestions.

with a great regards

---

## [Editor Report · Decision Letter 1]

23 Jul 2024

The Experience of Living with Human Immunodeficiency Virus among Adolescents at Felege Hiwot Comprehensive Specialized Hospital Bahir-Dar, Northwest Ethiopia, A Phenomenological Study

PONE-D-24-01002R1

Dear Dr. Bazezew Asfaw Guadie

We’re pleased to inform you that your manuscript has been judged scientifically suitable for publication and will be formally accepted for publication once it meets all outstanding technical requirements.

Kind regards,

Sogo France Matlala, PhD

Academic Editor

PLOS ONE
---

## [Editor Report · Acceptance letter]

8 Nov 2024

PONE-D-24-01002R1 

PLOS ONE

Dear Dr. Guadie, 

I'm pleased to inform you that your manuscript has been deemed suitable for publication in PLOS ONE. Congratulations! Your manuscript is now being handed over to our production team.

Kind regards, 

on behalf of

Professor Sogo France Matlala 

Academic Editor

PLOS ONE